# 🦫 Ferret-v2: An Improved Baseline for Referring and Grounding with Large Language Models

**Haotian Zhang[1]\*, Haoxuan You[2]\*, Philipp Dufter[1], Bowen Zhang[1], Chen Chen[1],
Hong-You Chen[1], Tsu-Jui Fu[3], William Yang Wang[3], Shih-Fu Chang[2], Zhe Gan[1], Yinfei Yang[1]**
[1]Apple AI/ML, [2]Columbia University, [3]UC Santa Barbara
{haotian_zhang2,zhe.gan,yinfeiy}@apple.com, haoxuan.you@cs.columbia.edu, tsu-juifu@ucsb.edu

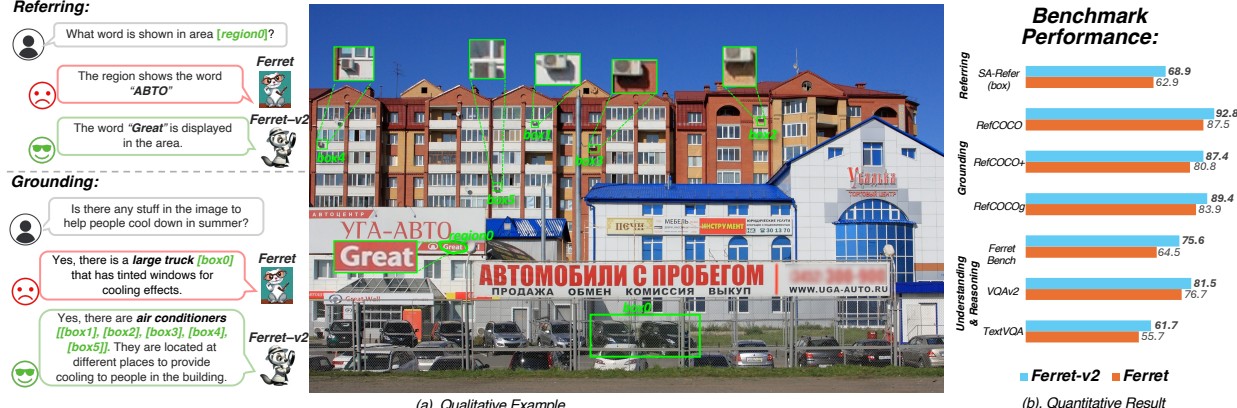

Figure 1: (a) The comparison showcases Ferret-v2's superior referring and grounding abilities over Ferret, particularly in identifying objects and texts within small regions. (b) Ferret-v2 notably exceeds Ferret's performance in tasks requiring detailed regional and global reasoning and understanding (all w/ 7B models).

## Abstract

While Ferret seamlessly integrates regional understanding into the Large Language Model (LLM) to facilitate its referring and grounding capability, it poses certain limitations: constrained by the pre-trained fixed visual encoder and failed to perform well on broader tasks. In this work, we unveil Ferret-v2, a significant upgrade to Ferret, with three key designs. (1) Any resolution grounding and referring: A flexible approach that effortlessly handles higher image resolution, improving the model's ability to process and understand images in greater detail. (2) Multi-granularity visual encoding: By integrating the additional DINOv2 encoder, the model learns better and diverse underlying contexts for global and fine-grained visual information. (3) A three-stage training paradigm: Besides image-caption alignment, an additional stage is proposed for high-resolution dense alignment before the final instruction tuning. Experiments show that Ferret-v2 provides substantial improvements over Ferret and other state-of-the-art methods, thanks to its high-resolution scaling and fine-grained visual processing.

## 1 Introduction

Multimodal Large Language Models (MLLMs) (Koh et al., 2023; Wu et al., 2023a; Yang et al., 2023; Liu et al., 2023d; Wu et al., 2023c; Li et al., 2023d; Ye et al., 2023; Wu et al., 2023b; Li et al., 2023a; Wang et al., 2023a; Gao et al., 2024; McKinzie et al., 2024) have increasingly become pivotal in the recent surge of advancements in AI, serving as foundational elements

---

[†]Equal contribution.

in the development of versatile general-purpose assistants. However, these methods were built on coarse image-level alignments, which suffer from fine-grained understanding (such as region description and reasoning). To this end, Peng et al. (2023); Chen et al. (2023b); You et al. (2023) integrate the grounding abilities and unlock the referential ability in dialogue, *i.e.*, enable the user to point to the object or region as input, and the model response with spatial coordinates of bounding boxes. Such capability fulfills fine-grained vision tasks, which is great progress in MLLMs.

While grounding and referring MLLMs exhibit strong performance, there are still many challenges that remain unresolved. For example, the aforementioned methods use CLIP (Jiang et al., 2023) or its variants (Sun et al., 2023) as the vision encoder. As the pre-trained image encoders normally adopt a relatively low image resolution, *e.g.*, 224×224, it severely hinders fine-grained visual comprehension for MLLMs. Though some task-specific MLLMs (Lv et al., 2023; Hong et al., 2023; Ye et al., 2023) have explored strategies for upscale processing, these approaches are marred by undue complexity for their own domains and cannot perform well on traditional MLLM benchmarks. Thus, the scenario prompts a critical inquiry: *how can we enhance the capabilities of MLLMs to excel in detailed vision-related tasks without compromising their proficiency in global reasoning?*

To answer this question, we explore the potential from three aspects, *i.e.*, higher-resolution scaling, multi-granularity visual encoding, and model training recipes. We choose Ferret (You et al., 2023) as the robust baseline since it has two advantages: (i) mutual benefits between referring and grounding, and (ii) more versatile referring capability (strokes, scribbles, or complex polygons). Firstly, we conduct a careful investigation into higher-resolution scaling, and evaluate the performance of two mainstream methods, "direct upsampling" (Wang et al., 2023a; Bai et al., 2023) and "any resolution" (Gao et al., 2024; Liu et al., 2024), on (i) Visual detail analysis (ROC (You et al., 2023) & REC (Kazemzadeh et al., 2014)), (ii) Resolution-critical OCR tasks (TextVQA (Singh et al., 2019)), and (iii) Reasoning MLLM benchmarks (Ferret-Bench (You et al., 2023)). Our analysis indicates that the "any resolution" approach outperforms "direct upsampling" in harnessing image details while retaining the knowledge acquired during pre-training for efficient scaling. This positions "any resolution" as a superior strategy for tasks requiring advanced visual comprehension.

By adopting the "any resolution" method, which involves dividing the image into sub-patches for processing by the CLIP encoder, we observed that incorporating both global context and high-resolution patches into visual embeddings introduces a nuanced complexity. This is because two types of images exhibits distinct characteristics. To mitigate this gap, we propose the integration of a DINOv2 encoder (Oquab et al., 2023). Renowned for its proficiency in delineating finer details pertaining to local objects, DINOv2 promises to bolster the model's ability to perceive fine-grained aspects. Additionally, we employ separate MLP projectors for each vision encoder to facilitate a deeper exploration of the varying contexts presented by global and fine-grained visual information, aiming for a more comprehensive understanding and representation.

Furthermore, the model is strategically trained in three stages, enhancing resolution handling while maintaining vision-language alignment in a "coarse-to-fine" manner. Initially, the model is trained on low-resolution images for efficient image-caption alignment. Subsequently, we recognize the gap that several downstream tasks demand a more accurate and thorough spatial understanding and go beyond just the broad semantics, so we specifically design the 2nd stage to align every possible local object of the image with detailed semantics with dense referring and detection data. Finally, the model undergoes visual instruction fine-tuning to better interpret user intent.

The contributions of this paper are summarized as follows: *(i)* We provide a thorough analysis of higher-resolution scaling, and found that the "any resolution" method consistently outperforms "direct upsampling". *(ii)* Based on "any resolution", we further propose multi-granularity visual encoding, where the low-resolution image is encoded via CLIP, while the high-resolution sub-patches are encoded via DINOv2. This strategy fosters a deeper understanding of both global and fine-grained visual contexts. *(iii)* Ferret-v2 is trained in a three-stage process, where an additional stage is proposed for high-resolution dense alignment before the final instruction tuning. Extensive experiments on a wide range

of tasks, including referring and grounding, visual question answering, and modern MLLM benchmarks demonstrate the superiority of Ferret-v2 over existing works (see Fig. 1).

## 2 Background

**Coarse-level MLLMs.** Motivated by the advanced reasoning abilities demonstrated by LLMs (OpenAI, 2022; Chowdhery et al., 2022; Touvron et al., 2023a;b; Zhang et al., 2022b; Wei et al., 2021), there is a growing interest in extending these skills to visual understanding, leading to the emergence of multimodal LLMs. For example, Flamingo (Alayrac et al., 2022) utilizes a cross-attention mechanism to enhance visual context awareness, enabling more sophisticated context-aware visual learning. Models such as LLaVA (Liu et al., 2023b;a) and MiniGPT-4 (Zhu et al., 2023) focus on synchronizing image and text features before applying instruction tuning. Additionally, BLIP-2 (Li et al., 2023d) and mPLUG-OWL (Ye et al., 2023) offer methods for incorporating image features using a visual encoder, which is then combined with textual embeddings in the LLM architecture. Nonetheless, despite their advancements, these MLLMs, including the latest GPT-4V (OpenAI, 2023), are limited to producing text outputs, restricting their application in scenarios that demand rich region-level visual perception.

**Region-level MLLMs.** In recent investigations, there has been a growing focus on the convergence of foundation models and the tasks related to dense visual perception. For example, Li et al. (2023c); Zou et al. (2023); Koh et al. (2023) leverage the CLIP pre-trained foundation models to enable open-world detection, but they are unable to handle complex instructions. Differently, VisionLLM (Wang et al., 2023c) combines a range of vision-centric tasks by utilizing instruction tuning with LLMs. However, it may fall short of fully harnessing the potential of LLMs for handling intricate reasoning tasks. In parallel research efforts, grounding capabilities and open-vocabularies detectors are leveraged by Kosmos-2 (Peng et al., 2023), Qwen-VL (Bai et al., 2023) and DetGPT (Pi et al., 2023), enabling user-guided detection. Moreover, GPT4RoI (Zhang et al., 2023b), Shikra (Chen et al., 2023b), LLaVA-G (Zhang et al., 2023a), and Ferret (You et al., 2023) introduce spatial boxes as input and train the model using region-text pairs, offering regional image understanding. However, all the above methods utilize low-resolution image encoders and thus limit the capability of perceiving more detailed analysis.

## 3 Methods

We first revisit the design principles of Ferret in Sec. 3.1 and present the investigation into higher-resolution scaling in Sec. 3.2. Subsequently, in Sec. 3.3, we delve into advancements in the model architecture, including techniques for grounding and referring at any resolution, as well as visual encoding with multiple granularities. Finally, we introduce an enhanced training methodology aimed at refining the model's proficiency in aligning global and local elements in Sec. 3.4.

### 3.1 A Revisit of Ferret

In recent investigations, there has been a growing focus on the convergence of models (Zhang et al., 2023b; Chen et al., 2023b; Peng et al., 2023; Lai et al., 2023; Zhao et al., 2023; You et al., 2023) and the tasks related to visual perception. Ferret (You et al., 2023) distinguishes itself from other MLLMs by excelling in spatial referring and grounding within natural images of diverse shapes and levels of detail.

To refer to various types of regions, such as points, boxes, or free-form shapes, Ferret developed a hybrid region representation, where each region is referred to by a combination of discrete coordinate tokens and continuous region features, as well as region names if available. The coordinates are normalized into the range from 0 to 1000, and a point or shape is respectively expressed by $[x, y]$ or $[x_{\min}, y_{\min}, x_{\max}, y_{\max}]$. The continuous region feature is extracted by a spatial-aware visual sampler that samples and aggregates features of the region. Ultimately, a region is represented by "⟨region_name⟩ ⟨coordinates⟩

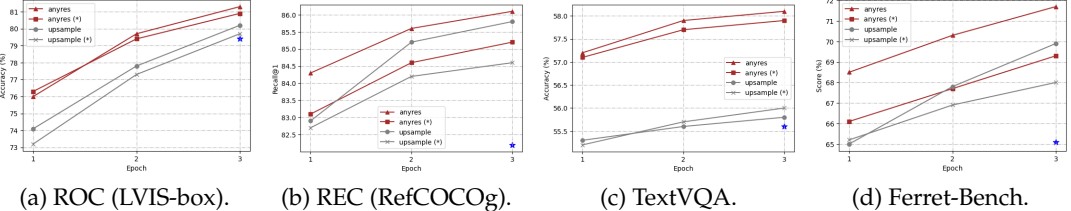

| (a) ROC (LVIS-box). | (b) REC (RefCOCOg). | (c) TextVQA. | (d) Ferret-Bench. |

Figure 2: Performance of "direct upsampling" and "any resolution" w/ 448×448 image resolution in ROC, REC, TextVQA, and Ferret-Bench. (∗ indicates the encoder is frozen during fine-tuning. ⋆ is denoted as vanilla Ferret w/ image resolution of 336×336.)

⟨continuous_fea⟩" and fed into the model for referring. e.g., "What is in the region [100, 50, 200, 300] ⟨continuous_fea⟩?". To achieve grounding, Ferret generates the box coordinates right after the corresponding regions/nouns in the text response, e.g., "There is a dog [100, 150, 300, 200] in the figure."

Ferret encodes the image with a pre-trained visual encoder (CLIP-ViT-L/14) (Radford et al., 2021) and then feeds the image feature as additional tokens alongside the text input (and hybrid region representation if any) into a decoder-only language model (Vicuna (Zheng et al., 2023)). The training contains two stages, image-caption alignment pre-training and instruction-tuning, updated with next-token-prediction loss.

While Ferret boasts flexibility and superior performance, it is hindered by the limitations imposed by the fixed resolution of its pre-trained encoder, which restricts its ability to fully exploit the advantages of enhanced region referring and localization accuracy. Motivated by this, we initially delve into identifying the most efficacious methods for high-resolution scaling. Subsequently, we unveil Ferret-v2, a substantial extension of the Ferret series, aimed at examining a broader and more inclusive multimodal learning framework.

## 3.2 Analysis of Higher Resolution Scaling

For further analysis, we conduct a series of controlled experiments using different high-resolution scaling methods, *i.e.*, "direct upsampling", and "any resolution"(Liu et al., 2024). The overall architecture and training process follows Ferret (You et al., 2023) but with a simple modification from a linear layer to a two-layer Multi-Layer Perceptrons (MLPs). Additionally, to enable the model to better handle short-form answers and perform on more benchmarks, we follow LLaVA 1.5 (Liu et al., 2023b) and add additional task-oriented datasets for VQA (Antol et al., 2015) and OCR to the existing GRIT (You et al., 2023), which was previously used in Ferret. To streamline our study, we choose 4 representative tasks: ROC (LVIS: box), REC (RefCOCOg), TextVQA, and Ferret-Bench, and measure the capability of the trained models comprehensively.

**Direct upsampling *v.s.* Any resolution.** For uniformity in our experiment, we standardize on a target resolution of 448 [1], which is upscaled from 336 as the vision encoder's pre-training resolution for both scaling methods to ensure identical image tokens are input into the LLMs. In the case of "direct upsampling", positional embedding interpolation is applied, and the CLIP backbone is adjusted to this new resolution during the fine-tuning phase. For "any resolution", we predefined a set of resolutions to support up to six grids [2]. Given a image, we first select the optimal resolution by prioritizing fitting the original image's aspect ratio and size as closely as possible while minimizing wasted resolution, and we resize the input image to the optimal resolution and split the image into these grids. All image patches are encoded by the CLIP encoder separately, and their features are input into

---

[1]The numbers of tokens are dynamic given different input image resolutions, but the maximum number of tokens is 1280. We chose 448 with computational overhead in mind.

[2]We use grid configurations of {1x1, 1x2, 1x3, 1x4, 1x5, 1x6, 2x2, 2x3, and their transpose}

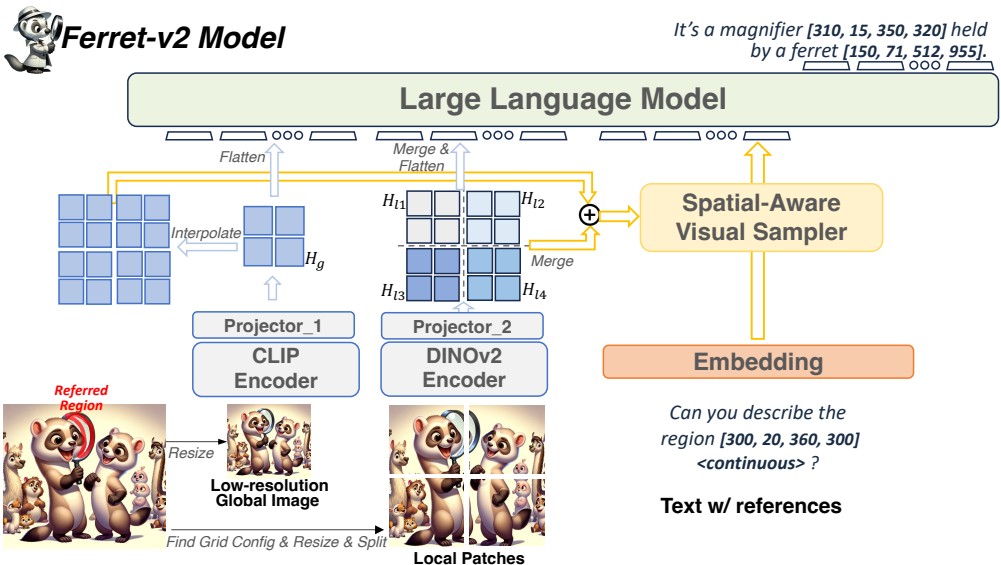

Figure 3: Overview of the proposed Ferret-v2 model architecture.

LLMs as image tokens. We trained the models using both frozen and unfrozen encoder configurations.

As highlighted in Fig. 2, our comparative analysis revealed that the "any resolution" scaling method not only demonstrated significant improvements across all tasks over the vanilla Ferret but also outshined the "direct upsampling" approach. Another interesting observation is that in "any resolution", updating the vision encoder always brings a boost over freezing it, whereas in "direct upsampling", freezing the vision encoder is sometimes even better (as shown in the TextVQA result). As for the reason behind those findings, we hypothesize that "direct upsampling" forces the ViT to adapt to a higher resolution, which brings much longer token lengths deviated from its pre-training data. However, the scale of fine-tuning data is usually much smaller than the pre-training data of the vision encoder (1.3M vs. 400M in our setting), which disturbs its pre-training knowledge. On the contrary, "any resolution" crops the high-resolution image into patches, and the vision encoder processes local patches in a similar token length to its pre-training procedure. Overall, "any resolution" has proved to be a more optimal strategy that balances leveraging high-resolution images and preserving valuable pre-training knowledge for effective scaling.

### 3.3 Model Architecture

**Multi-Granularity Visual Encoding.** With the adoption of the "any resolution" scaling method, yet another problem arises naturally: there is a granularity difference between global low-resolution image $I_g$ and local split image patches $\{I_{l1}, I_{l2}, ..., I_{lN}\}$, i.e., the global image $I_g$ sees the entire scene but in a coarse resolution, while each local patch $I_{li}$ can see only a part of the scene but in precise detail.

To deal with this issue, we explore encoding those two types of images with distinct visual encoders. Specifically, we choose the CLIP (Radford et al., 2021) to encode global images and DINOv2 (Oquab et al., 2023) to encode local split patches. Our motivation behind this comes from the difference in their pre-training paradigms. The image-text contrastive objective used in CLIP, enables these models to capture image-level semantics from captions but tends to neglect the rich pixel-level details due to the limited fine-grained information in the guided captions. DINOv2, trained with self-supervision objectives of both image-level and patch-level, can capture more detailed information about local objects such as shape or texture and therefore possess fine-grained perception abilities. Furthermore, we employ separate MLP projectors for the dual vision encoders, aiming to differentiate and learn the

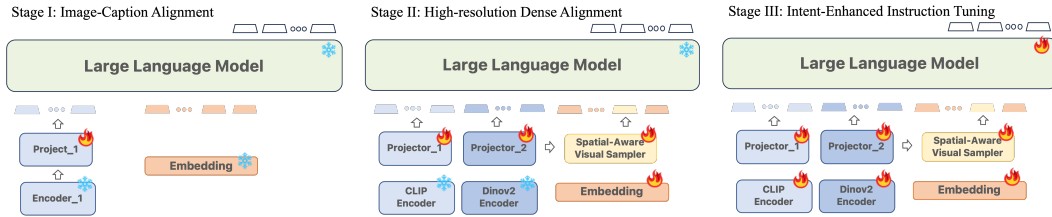

Figure 4: Model Training Paradigm. The model is trained from a "coarse-to-fine" manner. 'snowflake' denotes that the module is frozen.

diverse underlying contexts for global and fine-grained visual information:

$$F_g = \text{CLIP}(I_g); \qquad F_{li} = \text{DINO}(I_{li}), \qquad\qquad I_{li} \in \{I_{l1}, I_{l2}, ..., I_{lN}\} \qquad (1)$$

$$H_g = \text{MLP}_g(F_g); \quad H_{li} = \text{MLP}_l(F_{li}). \qquad\qquad (2)$$

Then, the feature maps of local patches are merged into a large feature map according to its original arrangement and then flattened into a sequence of image features. The global image's feature map is also flattened. Two sequences are connected and input into LLM as visual "tokens".

**Any resolution Referring.** The hybrid region representation introduced in Ferret has proved effective and versatile in handling various types of referring such as point, box, scribble, etc. What lies at the core of it is the extraction of continuous region features, which is performed by a Spatial-Aware Visual Sampler. However, directly feeding global image features into the visual sampler may not be sufficient to recognize the small referred objects in high-resolution images. Inspired by our previous findings about the visual granularity difference, we further propose to integrate the best of both global semantics and local details for more precise referring. To be more specific, after obtaining the encoded features of global image $H_g$ and local patches $\{H_{l1}, H_{l2}, ..., H_{lN}\}$, we first merge the feature maps of local patches into a large feature map following their original spatial arrangement, and the global image feature map is upsampled via interpolation to align the size of the merged feature map.

$$H_l' = \text{Concat}\{H_{l1}, H_{l2}, ..., H_{lN}\} \qquad (H_{li} \in \mathbb{R}^{w_l \times h_l \times c}, H_l' \in \mathbb{R}^{nw_l \times mh_l \times c}, n \times m = N) \qquad (3)$$

$$H_g' = \text{Upsample}(H_g) \qquad\qquad (H_g \in \mathbb{R}^{w_g \times h_g \times c}, H_g' \in \mathbb{R}^{nw_l \times mh_l \times c}) \qquad (4)$$

Then, we fuse the two processed feature maps by adding them channel-wise: $H_a = H_l' + H_g'$, and obtain a high-resolution feature map with strong semantics and local awareness. The $H_a$ is input into a spatial-aware visual sampler (You et al., 2023) to extract continuous region features. Then the continuous feature is combined with discrete coordinates as a hybrid region representation to refer to any region in the image, as shown in Fig. 3.

**Any resolution Grounding.** By combining visual embeddings from both global image and local sub-patches, our model can more effectively uncover visual details from high resolution and bridge the semantics. Without specific adaptation, our framework aligns seamlessly with the grounding design in Ferret; therefore, similarly, we delineate the output coordinate regions through an intuitive numerical representation and employ the LLM as the principal mechanism for deciphering the intrinsic correlations.

## 3.4 Training Paradigm

**Stage I: Image-Caption Alignment.** Feature alignment before fine-tuning has been widely utilized to achieve better training efficiency. We adopt this strategy to connect the pre-trained CLIP encoder with the LLM using 1.4M image-text pairs, converted to instruction-following data by (Chen et al., 2023c). The low-resolution image encoder and LLM parameters remain frozen, with only the projector trainable. Without any referring in these image-text pairs, the visual sampler doesn't participate in the training of stage I.

**Stage II: High-resolution Dense Alignment.** Although the previous image-caption alignment pre-training is effective in bridging vision and LLM in coarse semantics, there still exists a severe gap between the image-caption alignment and instruction tuning stage. Many downstream tasks, such as referring, grounding, OCR, etc, require a more precise and comprehensive spatial perception of the image, beyond solely coarse semantics.

To alleviate the above-mentioned issue, we propose a novel pre-training stage aimed at high-resolution dense alignment. Specifically, instead of aligning the entire image with a global caption, this stage aligns every possible local object of the image with detailed semantics. Correspondingly, two types of tasks and input data are designed. (1) ***Dense Referring***: given the image, the input question refers to regions of all objects one by one, and asks about their categories, the model is required to output the predicted classes accordingly. An example is "***Question:*** *Please classify the objects in the following locations. 1:* ⟨*region_1*⟩*, 2:* ⟨*region_2*⟩*, ....* ***Answer:*** *Here are the categories: 1: cat, 2: dog, ...*". (2) ***Dense Detection:*** Given the image, the input question asks to localize all the objects. To reduce randomness and incorporate spatial awareness, we forge the answer to list objects in a certain order, such as raster scan order (from top to bottom, from left to right). An example is "***Question:*** *Please localize visible objects in the image in a raster scan order.* ***Answer:*** *The objects are: 1: cat* ⟨*coordinate_1*⟩*, 2: dog* ⟨*coordinate_2*⟩*, ...*". To ensure the efficient learning of the fine-grained semantics, we collect data from densely annotated object dataset - LVIS (Gupta et al., 2019). On average, each sample includes around 10 object locations, whereas in the instruction tuning stage, referring and grounding datasets mostly have only one or two object locations mentioned per sample.

In terms of the model, we take a pre-trained DINOv2 as the visual encoder for local patches, in addition to the CLIP encoder for global images, as mentioned in Sec. 3.3. The projector after CLIP is inherited from the image-caption alignment stage, and we further add a separate projector after DINOv2, whose weights are initialized from the CLIP's projector for stability. Then we freeze two vision encoders and LLMs, and only update the two projectors as well as the visual sampler in this alignment stage, with the next-token-prediction loss.

**Stage III: Intent-Enhanced Instruction Tuning.** After the second stage of pre-training, the model acquires the capability for a comprehensive global understanding of images, alongside the ability to identify and narrate objects of interest using free-form texts and visually referred regions obtained flexibly. Our aim is to enhance the model's adherence to user instructions while maintaining its high-resolution visual perception abilities. To achieve this, we render the encoders, projectors, region samplers, and the LLM itself trainable. For training, we utilize the GRIT dataset (You et al., 2023) and incorporate additional task-specific datasets for VQA (Antol et al., 2015) and OCR (Singh et al., 2019; Sidorov et al., 2020) from LLaVA 1.5 (Liu et al., 2023b). Furthermore, we identified two additional strategies that contribute to enhanced performance: (i) Data Unification: To facilitate the model's seamless transition from a global understanding based on plain texts to a regional comprehension utilizing hybrid representations, we employ an open-vocabulary object detector, GLIPv2 (Zhang et al., 2022a), to localize groundable nouns in the text on VQA datasets, and a public OCR model (Kuang et al., 2021) to get text bounding boxes on OCR datasets. (ii) Task Generalization: In order to diminish ambiguity across tasks that necessitate referring and grounding capabilities and those that do not, we adopt a method similar to LLaVA 1.5, which involves appending the prompt, "*Include the coordinates for each mentioned object.*", to further clarify task requirements.

## 4   Experiments

### 4.1   Referring and Grounding Tasks

**Referring.** Ferret-v2's enhanced understanding of referential queries is evident in its ability to interpret the semantics of specified regions within an image accurately. This is particularly assessed through the task of Referring Object Classification (ROC), where the model is tasked with identifying the object in a region mentioned in a query. Initially, like Ferret, we utilize the validation split of the LVIS dataset, covering more than 1,000 object

| Models | LVIS (%) | | | SA-refer (%) | | |
|---|---|---|---|---|---|---|
| | Point | Box | Free-form | Point | Box | Free-form |
| Random Guess | 50 | 50 | 50 | 50 | 50 | 50 |
| Kosmos-2 | ✗ | 60.25 | ✗ | ✗ | 53.97 | ✗ |
| Shikra-7B | 57.82 | 67.71 | ✗ | 54.15 | 56.82 | ✗ |
| GPT4-ROI | ✗ | 61.76 | ✗ | ✗ | 55.02 | ✗ |
| CogVLM-17B | ✗ | 79.62 | ✗ | ✗ | 61.77 | ✗ |
| SPHINX-2k | 72.83 | 82.97 | ✗ | 61.21 | 63.39 | ✗ |
| Ferret-7B | 67.94 | 79.42 | 69.77 | 61.91 | 62.99 | 57.74 |
| Ferret-v2-7B (Ours) | 74.55 | 86.59 | 76.13 | 68.38 | 68.83 | 62.07 |
| Ferret-13B | 68.35 | 80.46 | 70.98 | 63.16 | 63.35 | 58.02 |
| Ferret-v2-13B (Ours) | 75.09 | 87.74 | 76.35 | 67.38 | 69.49 | 62.58 |

Table 1: Results of ROC on three different referring types, including point, box, and free-form shape. '✗' means no such capability.

| Models | Ferret-Bench | | | |
|---|---|---|---|---|
| | Referring Description | Referring Reasoning | Grounding in Conversation | Avg. |
| LLaVA | 41.4 | 31.7 | 28.8 | 34.0 |
| Kosmos-2 | 51.8 | 33.7 | 48.4 | 44.6 |
| Shikra-7B | 46.0 | 41.6 | 50.1 | 45.9 |
| CogVLM-17B | 67.1 | 67.6 | 51.7 | 62.1 |
| Osprey-7B | 72.2 | 67.8 | – | – |
| SPHINX-2k | 55.6 | 70.2 | 66.4 | 64.0 |
| Ferret-7B | 68.7 | 67.3 | 57.5 | 64.5 |
| Ferret-v2-7B (Ours) | 79.9 | 81.7 | 65.2 | 75.6 |
| Ferret-13B | 70.6 | 68.7 | 59.7 | 66.3 |
| Ferret-v2-13B (Ours) | 79.6 | 79.4 | 65.7 | 74.9 |

Table 2: Results on the proposed Ferret-Bench via GPT4-as-a-Judge evaluation.

| Models | RefCOCO | | | RefCOCO+ | | | RefCOCOg | | Flickr30k Entities | |
|---|---|---|---|---|---|---|---|---|---|---|
| | val | testA | testB | val | testA | testB | val | test | val | test |
| MAttNet (Yu et al., 2018) | 76.40 | 80.43 | 69.28 | 64.93 | 70.26 | 56.00 | 66.67 | 67.01 | – | – |
| OFA-L (Wang et al., 2022) | 79.96 | 83.67 | 76.39 | 68.29 | 76.00 | 61.75 | 67.57 | 67.58 | – | – |
| UNITER (Chen et al., 2020) | 81.41 | 87.04 | 74.17 | 75.90 | 81.45 | 66.70 | 74.02 | 68.67 | – | – |
| VILLA (Gan et al., 2020) | 82.39 | 87.48 | 74.84 | 76.17 | 81.54 | 66.84 | 76.18 | 76.71 | – | – |
| UniTAB (Yang et al., 2022) | 86.32 | 88.84 | 80.61 | 78.70 | 83.22 | 69.48 | 79.96 | 79.97 | 78.76 | 79.58 |
| MDETR (Kamath et al., 2021) | 86.75 | 89.58 | 81.41 | 79.52 | 84.09 | 70.62 | 81.64 | 80.89 | 82.3* | 83.8* |
| G-DINO-L (Liu et al., 2023c) | 90.56* | 93.19* | 88.24* | 82.75* | 88.95* | 75.92* | 86.13* | 87.02* | – | – |
| Shikra-7B (Chen et al., 2023b) | 87.01 | 90.61 | 80.24 | 81.60 | 87.36 | 72.12 | 82.27 | 82.19 | 75.84 | 76.54 |
| MiniGPT-v2-7B (Chen et al., 2023a) | 88.06 | 91.29 | 84.30 | 79.58 | 85.52 | 73.32 | 84.19 | 84.31 | – | – |
| Qwen-VL-7B (Bai et al., 2023) | 88.55 | 92.27 | 84.51 | 82.82 | 88.59 | 76.79 | 85.96 | 86.32 | – | – |
| SPHINX-2k (Lin et al., 2023) | 91.10 | 92.88 | 87.07 | 85.51 | 90.62 | 80.45 | 88.07 | 88.65 | – | – |
| LLaVA-G (Zhang et al., 2023a) | 89.16 | – | – | 81.68 | – | – | 84.82 | – | 83.03 | 83.62 |
| VistaLLM (Pramanick et al., 2023) | 88.1 | 91.5 | 83.0 | 82.9 | 89.8 | 74.8 | 83.6 | 84.4 | – | – |
| Ferret-7B (You et al., 2023) | 87.49 | 91.35 | 82.45 | 80.78 | 87.38 | 73.14 | 83.93 | 84.76 | 80.39 | 82.21 |
| Ferret-v2-7B (Ours) | 92.79 | 94.68 | 88.69 | 87.35 | 92.75 | 79.3 | 89.42 | 89.27 | 85.52 | 85.83 |
| Shikra-13B (Chen et al., 2023b) | 87.83 | 91.11 | 81.81 | 82.89 | 87.79 | 74.41 | 82.64 | 83.16 | 77.41 | 78.44 |
| Griffon v2 (Zhan et al., 2024) | 89.6 | 91.8 | 86.5 | 81.9 | 85.5 | 76.2 | 85.9 | 86.0 | – | 84.8 |
| CogVLM-Grounding-17B (Wang et al., 2023a) | 92.76 | 94.75 | 88.99 | 88.68 | 92.91 | 83.39 | 89.75 | 90.79 | – | – |
| Ferret-13B (You et al., 2023) | 89.48 | 92.41 | 84.36 | 82.81 | 88.14 | 75.17 | 85.83 | 86.34 | 81.13 | 84.76 |
| Ferret-v2-13B (Ours) | 92.64 | 94.95 | 88.86 | 87.39 | 92.05 | 81.36 | 89.43 | 89.99 | 85.33 | 86.25 |

Table 3: Performance comparison (Acc@0.5) on the REC (RefCOCO, RefCOCO+, RefCOCOg) and phrase grounding (Flickr30k Entities) tasks. * indicates that the method is specifically fine-tuned in the second stage.

categories with a majority being "in-domain" images. To further demonstrate Ferret-v2's improved ability to reference smaller objects, we compile an "in-the-wild" evaluation set using partial images from SA-1B (Kirillov et al., 2023) and corresponding human annotations of objects from AS-human (Wang et al., 2023b), which contains high-resolution images, open-vocabulary objects and precise masks. In total, we manually verified 700+ high-quality samples with in-the-wild objects and called it SA-refer. As shown in Table 1, Ferret-v2 significantly outperforms previous models on LVIS and sets up a new benchmark not fully realized in prior Ferret, primarily contributing to high-resolution scaling. SPHINX also uses high-resolution input images; however, on more challenging tasks for SA-1B, Ferret-v2 still outperforms it, indicating the benefits of our special design for any resolution referring.

**Grounding.** Visual grounding aims to ground language queries into aligned image regions. We experiment on the sub-tasks of referring expression comprehension (REC) with three renowned benchmarks: RefCOCO (Lin et al., 2014), RefCOCO+ (Yu et al., 2016), and RefCOCOg (Mao et al., 2016), and phrase grounding with Flickr30k Entities dataset (Plummer et al., 2015). As evidenced in Table 3, Ferret-v2 enables the use of high-resolution input images, leading to significant improvements over Ferret (You et al., 2023). Besides, Ferret-v2 outperforms most state-of-the-art models, including specialist model G-DINO-L (Liu et al., 2023c) and other generalist models, which adopt even larger input image sizes. Our 7B model can achieve comparable results to CogVLM-Grounding (Wang et al., 2023a), which utilizes a 4B vision model and a 6B connection module. These results demonstrate the competitive capability of Ferret-v2 for visual grounding.

**Ferret-Bench.** Ferret-Bench (You et al., 2023) is carefully designed to evaluate and benchmark the fine-grained capability of multimodal conversational models, particularly in their

| Method | VQA$^{v2}$ | GQA | VQA$^T$ | POPE | MME$^P$ | SEED | LLaVA$^C$ | LLaVA$^W$ | MM-Vet | Obj-Hal ↓ |
|---|---|---|---|---|---|---|---|---|---|---|
| BLIP-2-13B | 41.0 | 41 | 42.5 | 85.3 | 1293.8 | 46.4 | – | 38.1 | 22.4 | – |
| InstructBLIP-7B | – | 49.2 | 50.1 | – | – | 53.4 | – | 60.9 | 26.2 | – |
| IDEFICS-9B | 50.9 | 38.4 | 25.9 | – | – | – | – | – | – | – |
| Qwen-VL-7B | 78.8* | 59.3* | 63.8 | – | – | 56.3 | – | – | – | – |
| Qwen-VL-Chat-7B | 78.2* | 57.5* | 61.5 | – | 1487.5 | 58.2 | – | – | – | 43.8/23.0 |
| LLaVA-1.5-7B | 78.5* | 62.0* | 58.2 | 85.9 | 1510.7 | 58.6 | 82.7 | 63.4 | 30.5 | 46.3/22.6 |
| Ferret-v2-7B (Ours) | **81.5*** | **64.7*** | 61.7 | 87.8 | 1510.3 | 58.7 | 89.1 | 67.7 | 34.9 | **23.8/14.7** |
| InstructBLIP-13B | – | 49.5 | 50.7 | 78.9 | 1212.8 | – | – | 58.2 | 25.6 | – |
| Shikra-13B | 77.4* | – | – | – | – | – | – | – | – | – |
| IDEFICS-80B | 60.0 | 45.2 | 30.9 | – | – | – | – | – | – | – |
| LLaVA-1.5-13B | 80.0* | 63.3* | 61.3 | 85.9 | 1531.3 | 61.6 | 83.4 | 70.7 | 35.4 | – |
| LLaVA-1.5-13B-HD | 81.8* | 64.7* | **62.5** | 86.3 | 1500.1 | **62.6** | – | **72.0** | **39.4** | – |
| Ferret-v2-13B (Ours) | 81.8* | **64.8*** | 62.2 | 88.1 | **1521.4** | 61.7 | 90.7 | 69.9 | 35.7 | 34.7/16.8 |

Table 4: Comparison with SoTA methods on 10 benchmarks. Ferret-v2 achieves comparable performance with others. *The training images of the datasets are observed during training.

| Resolution | Referring | | Grounding | OCR | Reasoning |
|---|---|---|---|---|---|
| | LVIS | SA | REC | TextVQA | Ferret-Bench |
| Fixed Res. | 68.4 | 61.9 | 86.8 | 54.2 | 71.1 |
| + AnyRes. Ground | 72.2 | 67.7 | 88.3 | 60.2 | 72.2 |
| + AnyRes. Refer | 73.0 | 67.8 | 88.5 | 60.7 | 72.2 |

Table 5: Ablation study on any resolution grounding and referring.

| Model | Referring | | Grounding | OCR | Reasoning |
|---|---|---|---|---|---|
| | LVIS | SA | REC | TextVQA | Ferret-Bench |
| CLIP | 73.0 | 67.8 | 88.5 | 60.7 | 72.6 |
| + DINOv2 | 73.8 | 68.0 | 89.1 | 61.3 | 75.3 |
| + Stage II | **74.6** | **68.4** | **89.3** | **61.7** | **75.6** |

Table 6: Ablation study on the effectiveness of the multi-granularity visual encoding and Stage II Pre-training.

ability to refer to, describe, and reason about specific regions within images, thereby facilitating a more structured evaluation of models' referring and grounding capabilities in a multimodal context. We use Ferret-Bench to compare Ferret with previous models, including LLaVA (Liu et al., 2023b), Shikra (Chen et al., 2023b), Kosmos-2 (Peng et al., 2023), and Osprey (Yuan et al., 2023). Results are summarized in Table 2. Ferret-v2 demonstrates superior performance in all types of tasks, indicating the strong spatial understanding and commonsense reasoning capability of the model.

## 4.2 Modern MLLM Benchmarks

In pioneering the fields of referring and grounding, Ferret has demonstrated remarkable region reasoning capabilities, as illustrated above. However, it falls short of academic benchmarks that typically demand tasks-oriented datasets. For Ferret-v2, we specifically include pseudo-labeled VQA and OCR datasets and also append the special prompt, as mentioned in Sec. 3.4. This strategic enhancement progressively narrows the gap between task-specific region-level analyses and broader, more generalized tasks, thereby extending Ferret-v2's applicability to encompass both fine-grained and coarse-grained tasks. As presented in Table 4, we benchmark Ferret-v2 against existing MMLMs across a comprehensive suite of 10 benchmarks: VQA$^{v2}$(Antol et al., 2015), TextVQA(aka.VQA$^T$) (Singh et al., 2019), GQA (Hudson & Manning, 2019), POPE (Li et al., 2023e), MME$^P$ (Chang et al., 2023), SEED (Li et al., 2023b), LLaVA$^C$ and LLaVA$^W$ (Liu et al., 2023b), MM-Vet (Yu et al., 2023b), Obj-Hal (Yu et al., 2023a)). Our models achieve on-par performance with the latest state-of-the-art models, particularly excelling in tasks such as VQAv2, GQA, POPE, etc., which demand precise spatial information for accurate responses.

## 5 Ablation Studies

In all the ablation studies below, we follow Sec. 3.2 and primarily focus our evaluation on the disparate models' performance across the dimensions of referring, grounding, OCR, and reasoning. Additionally, We also explore how Ferret-v2 balances accuracy with efficiency.

**Any Resolution Grounding and Referring.** We conduct an ablation study on any resolution grounding and referring. As illustrated in Table 5, accommodating any resolution

markedly enhances task performance that necessitates a comprehensive understanding of higher-resolution details. Integrating the best of both global semantics and local details for more precise and improved precision in referring tasks across both LVIS and SA datasets. Furthermore, this integration also modestly enhances grounding capabilities, suggesting that grounding and referring can derive mutual benefits within our proposed framework.

**Multi-Granularity Visual Encoding and Stage-II Pretrain.** Our initial ablation study focuses on incorporating an additional DINOv2 encoder for the encoding of high-resolution patches. We utilize the projector weights from Stage I of CLIP for initialization, followed by fine-tuning in Stage III. As demonstrated in Table 6, the exclusive employment of visual granularity encoding significantly enhances both referring and grounding performance. Furthermore, introducing an intermediary Stage II in the pre-training process yields improvements across all evaluated metrics.

**Efficiency and Scalability.** We add a detailed analysis of Ferret-v2's performance on Referring (LVIS) and Grounding (REC) and a comparison of the computational cost (in terms of inference time and average token per sec.) with the previous Ferret baseline.

In Table 7, we find that accommodating any resolution markedly enhances task performance, particularly for tasks that require a comprehensive understanding of higher-resolution details. Additionally, incorporating an extra DINOv2 encoder significantly improves referring and grounding performance. While this design leads to increased computational demands, the performance gains achieved by the model justify the additional computational costs.

| Design | LLM | Task | Performance | Avg. Inference Time (sec/it) | Avg. Tokens (per sec) |
|---|---|---|---|---|---|
| Base (Ferret) | Vicuna 1.3 | Referring (LVIS) Grounding (REC) | 67.9 83.9 | 1.4 0.4 | 6.1 5.3 |
| + Longer Context | Vicuna 1.5 | Referring Grounding | 68.4 86.8 | 1.1 0.3 | 4.8 5 |
| + AnyRes. Grounding | Vicuna 1.5 | Referring Grounding | 72.2 88.3 | 1.7 0.48 | 3.1 3.2 |
| + AnyRes. Referring | Vicuna 1.5 | Referring Grounding | 73 88.5 | 1.8 0.5 | 3 3.1 |
| + DINOv2 (Ferretv2) | Vicuna 1.5 | Referring Grounding | 73.8 89.1 | 2 0.54 | 2.8 2.9 |

Table 7: Performance comparison of different models and settings. All model inference are based on 7B scale and performed with 8 Nvidia A100 GPUs using greedy decoding.

## 6 Conclusions

We present Ferret-v2, a significant upgrade of the vanilla Ferret model. It features advanced capabilities in handling any resolution referring and grounding, multi-granularity visual encoding, and a novel three-stage training pipeline. These improvements enable Ferret-v2 to excel in processing and understanding images with higher resolution and finer detail.

## Limitations

Ferret-v2 mitigates harmful outputs by using transparent datasets, enhancing spatial knowledge to reduce object hallucinations, and improving instruction-following for unclear queries. However, its referring and grounding capabilities pose real-world challenges, particularly in interactive scenarios, where incorrect contextual understanding and ambiguous references could lead to inaccurate location outputs.

## Acknowledgment

The authors would like to thank Yizhe Zhang, Yanghao Li, Liangchen Song, and Keen You for valuable guidance, suggestions, and feedback. Additional thanks go to Jiaming Hu, Mingfei Gao for supporting large-scale training.

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
