# OpenReview forum: "Ferret-v2: An Improved Baseline for Referring and Grounding with Large Language Models"
_colmweb.org/COLM/2024/Conference — COLM_

### Official Review · Reviewer_aBcC · 2024-05-10

**Rating:** 6
**Confidence:** 4
**Ethics Flag:** 1

**Summary:**

The paper introduces Ferret-v2, an enhancement of the Ferret model, aimed at refining the capabilities of multimodal large language models (MLLMs) for high-resolution image understanding. The authors focus on improving the model's referring and grounding abilities, particularly in detailed regional and global reasoning. The introduction of a multi-granularity visual encoding and a three-stage training paradigm are novel and effective.  The results, compared against other state-of-the-art methods, demonstrate substantial improvements.

**Questions To Authors:**

(1) Could you elaborate on the potential limitations and ethical considerations associated with Ferret-v2, especially concerning its deployment in real-world applications?

(2) How much does the model training costs? Will the code and model publicly released?

**Reasons To Accept:**

(1) The paper is well-written, and clear, providing extensive empirical evidence to support its claims.

(2) The approach integrates innovative techniques like any resolution grounding, multi-granularity visual encoding, and a novel training paradigm, which are well justified and rigorously tested.

(3) The paper provides a comprehensive set of experiments and comparisons to benchmark datasets, demonstrating the effectiveness and efficiency of Ferret-v2.

**Reasons To Reject:**

(1) The paper briefly mentions potential risks of harmful and counterfactual responses from MLLMs but does not thoroughly explore limitations or potential biases of the proposed model.

(2) The complexity of the model and the training process might pose challenges for reproducibility, which are not adequately addressed in the manuscript.

---

> ### Author Rebuttal · Authors · 2024-05-31
>
> Thank you for your positive and constructive feedback on our paper. We appreciate your recognition of the clarity of our writing, the innovative techniques we introduced, and the comprehensive set of experiments provided. We address your concerns and questions below.
>
> **1. Ethical Considerations and potential risks for real-world deployment**
>
> Please refer to Response #2 for Reviewer NGY1 with shared ethical considerations. Ferret-v2 mitigates harmful outputs by using public, academic datasets for transparency and bias reduction, integrating fine-grained spatial knowledge to minimize object hallucinations, and incorporating instruction-following capabilities to handle unclear queries.
>
> Additionally, specific real-world deployment concerns arise from Ferret-v2's referring and grounding capabilities, especially in interactive scenarios where the model provides location instructions and interacts with users, including those with disabilities such as blindness or deafness. These concerns include:1) Contextual Understanding: Incorrect contextual interpretation can lead to erroneous location outputs. 2) Ambiguity in References: Users might provide ambiguous or unclear reference points/boxes/scribbles, which may result in inaccurate outputs. We will address these points in the revised paper.
>
> **2. Model training costs and Reproducibility**
>
> Ferret-v2 utilizes a multi-stage training paradigm. For Stage-II, Ferret-v2 is trained on densely partial annotated LVIS data (200k) for 1 epoch, optimized by AdamW with a learning rate of 1e-4 and a batch size of 256. we freeze two vision encoders and LLMs, and only update the two projectors as well as the visual sampler in this alignment stage and the training takes \~7 hrs on 8 40G A100 GPUs.
>
> For Stage III, Ferret-v2 is trained on aforementioned GRIT data and task-specific VQA and OCR data (~1.6M) for 3 epochs. optimized by AdamW with a learning rate of 2e-5 and a batch size of 256. In this stage, we render the encoders, projectors, region samplers, and the LLM itself trainable. The training takes ~20/27 hrs on 32 40G A100 GPUs for a Ferret-v2 7B/13B.
>
> We are committed to releasing the training/inference codes and model checkpoints for Ferret-v2. To comply with the double-blind policy, we created a **public anonymous repo** (https://anonymous.4open.science/r/ferret_v2_anno-85AF/). Upon our institution's approval, we will further organize the code and open-source the full training data and pre-trained model checkpoints.

---

> > ### Author Response · Authors · 2024-06-06
> > **Reviewer-Author Discussion Period Ends in ONE Day**
> >
> > Dear Reviewer aBcC,
> >
> > Thank you again for your insightful reviews of our submission. Following your feedback, we have provided a detailed response trying to address the concerns you raised. As the deadline is approaching, it would be very helpful if you could revisit our clarifications and let us know if any ambiguities remain before the reviewer-author discussion period ends. We would greatly appreciate any further comments you might have regarding our work, and we are fully committed to answering any questions.
> >
> > Your effort and time in reviewing our submission are sincerely appreciated.
> >
> > Warm regards,
> >
> > Author(s)

---

> > > ### Comment · Reviewer_aBcC · 2024-06-07
> > >
> > > Thank you for your reply. I have carefully read the response and I will keep my original ratings.

---

### Official Review · Reviewer_889e · 2024-05-10

**Rating:** 6
**Confidence:** 4
**Ethics Flag:** 1

**Summary:**

This paper is an improved version of Ferret. The improvement includes high image resolution, multi-granularity visual encoding and three-stage training pipeline. These techniques bring a lot benefit to Ferret and result to leading performance in various multimodal benchmarks,  especially on region-level tasks.

**Reasons To Accept:**

This paper introduces a series of effective techniques to improve the multimodal model,
- (1) high image resolution: sub-patches and direct upsampling are well studied.
- (2) multi-granularity visual encoding: taking advantages of CLIP and DINOv2 on global image feature extraction and low-level details preservation, respectively.
- (3) multi-stage training pipeline: I really appreciate that Stage III unfreezes CLIP and DINOv2, which is a very brave attempt and rarely seen in recent multimodal works.

By integrating these to Ferret, leading experimental results are produced in both image-level and region-level multimodal benchmarks.

**Reasons To Reject:**

The main concern is that this is an incremental paper from the perspective of technical originality. For example, the sub patches-based high resolution has been used in [1][2], the integration of CLIP and DINO as the vision encoder are studied in [3], the multi-stage training strategies are widely used in previous multimodal works.

This is the first COLM, the acceptance threshold is not clear to me. If technical originality is a highly strict requirement of COLM, Ferret-v2 may not be novel enough to be accepted. However, if a well-constructed engineering paper is recommended, Ferret-v2 is truly a good example.

My initial score is a conservative score. I will determine my final score after reading other reviewers comments.

Reference:
- [1] LLaVA-NeXT: Improved reasoning, OCR, and world knowledge
- [2] Towards Open-Ended Visual Recognition with Large Language Model
- [3] From CLIP to DINO: Visual Encoders Shout in Multi-modal Large Language Models

---

> ### Author Rebuttal · Authors · 2024-05-31
>
> Thank you for your detailed feedback on our paper. We appreciate your recognition of the effectiveness and performance of Ferret-v2. Below, we address your concerns and comments:
>
> **Incremental Technical Originality**
>
> While it is true that individual components have been touched separately in previous works, the novelty of Ferret-v2 lies in the unique design and combination of these techniques to enhance fine-grained multimodal performance comprehensively. Specifically:
>
> **a. Any resolution**: While previous works used “any resolution” strategies and “direct upsampling” for high-resolution images, our analysis across four tasks (ROC, REC, TextVQA, Ferret-Bench) shows “any resolution” optimally balances high-res image use with pre-training knowledge preservation, crucial for scaling.
> We designed Ferret-v2’s novel “referring” and “grounding” mechanisms. For referring, we integrate global semantics and local feature maps via a spatial-aware visual sampler. For grounding, our framework aligns with the LLM to delineate output regions.
>
> **b. Multi-granularity visual encoding**: We appreciate the reference to [3]. Unlike [3], which focuses on multi-level feature fusion of CLIP and DINOv2 features before feeding into LLMs, Ferret-v2 encodes global and sub-patch features separately with distinct encoders and then feeds them into LLMs simultaneously. This approach: 1) Treats global and local patches differently, respecting their unique characteristics. 2) Uses encoders' pre-training paradigms suited for global (CLIP) and local (DINOv2) patches. 3) Maintains separate tokens for global/local features, improving performance on referring and grounding tasks.
>
> **c. Multi-stage training**: We appreciate the reviewer’s acknowledgment of multi-stage pre-training trends. Our approach has distinct aspects compared to existing works, which often define training stages based on "dataset scales" without model changes.
> In contrast, our training includes a unique Stage II, using a "coarse-to-fine" approach by initializing two projectors and a visual sampler, setting the foundation for a nuanced strategy. Stage III unfreezes CLIP and DINOv2, "a bold move rarely seen in recent multimodal works " that the reviewer acknowledges and appreciates.
>
> Thank you again for your valuable feedback. We believe Ferret-v2 offers significant practical value and demonstrates satisfactory novelty, regardless of its extending series. We look forward to improving our paper based on your insights.

---

> > ### Author Response · Authors · 2024-06-06
> > **Reviewer-Author Discussion Period Ends in ONE Day**
> >
> > Dear Reviewer 889e,
> >
> > Thank you again for your insightful reviews of our submission. Following your feedback, we have provided a detailed response trying to address the concerns you raised. As the deadline is approaching, it would be very helpful if you could revisit our clarifications and let us know if any ambiguities remain before the reviewer-author discussion period ends. We would greatly appreciate any further comments you might have regarding our work, and we are fully committed to answering any questions.
> >
> > Your effort and time in reviewing our submission are sincerely appreciated.
> >
> > Warm regards,
> >
> > Author(s)

---

> > > ### Comment · Reviewer_889e · 2024-06-07
> > >
> > > I really appreciate the authors' response. I understand there are indeed technical differences between Ferret-v2 and previous works, but these differences are still incremental compared with Ferret-v1.
> > >
> > > After reading other reviewers' opinions and scores, I keep my score. Good luck !

---

### Official Review · Reviewer_NGY1 · 2024-05-11

**Rating:** 7
**Confidence:** 4
**Ethics Flag:** 1

**Summary:**

The authors propose Ferret-v2, an improved baseline of  Ferret. Ferret-v2 has three key designs: any resolution processing, multi-granularity visual encoding, and a three-stage training paradigm. The referring and grounding abilities of Ferret-v2 are strengthened by these designs. Finally, experiments demonstrated the effectiveness of the proposed Ferret-v2 on referring, grounding tasks, and modern MLLM benchmarks.

**Questions To Authors:**

please see the ''reasons to reject''

**Reasons To Accept:**

1. The authors propose an improved Ferret-v2 with three specialized designs to strengthen the referring and grounding capabilities. Ferret-v2 seems novel to me.
2. The proposed Ferret-v2 is effective and superior on multiple benchmarks, including referring tasks, grounding tasks, and modern MLLM benchmarks.
3. This paper is easy to follow.

**Reasons To Reject:**

1. Efficiency and scalability. How does Ferret-v2 balance accuracy with computational efficiency? The time complexity of Ferret-v2 is suggested to be clarified.
2. Potential harmful outputs and ethical considerations. The authors are suggested to discuss how Ferret-v2 prevents the potentially harmful outputs.

---

> ### Author Rebuttal · Authors · 2024-05-30
>
> Thank you for your thoughtful and constructive feedback on our paper. We are pleased that you found Ferret-v2 novel and effective and that you appreciated the clarity of our presentation. We address your concerns below.
>
> **1. Efficiency and scalability**
>
> We appreciate your concern regarding the efficiency and scalability of Ferret-v2. Here, we added a detailed analysis of Ferret-v2's performance on Referring (LVIS) and Grounding (REC) and a comparison of the computational cost (in terms of inference time and average token per sec) with the previous Ferret baseline. Additionally, we discuss the architectural choices that contribute to this balance. (*All model inference is based on a 7B scale and performed with 8 Nvidia A100 GPUs using greedy decoding.)
>
> | Design | LLM | Task | Performance | Average Inference Time (sec/it) | Average Token (per sec) |
> |-------------------------------|-------------|------------|-------------|---------------------------------|-------------------------|
> | Base (Ferret) | Vicuna 1.3 | Referring (LVIS) | 67.9 | 1.4 | 6.1 |
> | | | Grounding (REC) | 83.9 | 0.4 | 5.3 |
> | + longer context length (4k) | Vicuna 1.5 | Referring | 68.4 | 1.1 | 4.8 |
> | | | Grounding | 86.8 | 0.3 | 5 |
> | + AnyRes. Grounding | Vicuna 1.5 | Referring | 72.2 | 1.7 | 3.1 |
> | | | Grounding | 88.3 | 0.48 | 3.2 |
> | + AnyRes. Referring | Vicuna 1.5 | Referring | 73 | 1.8 | 3 |
> | | | Grounding | 88.5 | 0.5 | 3.1 |
> | + DINOv2 (Ferretv2 w/o Stage II) | Vicuna 1.5 | Referring | 73.8 | 2 | 2.8 |
> | | | Grounding | 89.1 | 0.54 | 2.9 |
>
> We find that "any resolution" and adding an extra DINOv2 encoder significantly improves performance. While this design leads to increased computational demands, the model's performance gains justify the additional computational costs. We will include this discussion in the revised paper.
>
> **2. Potential Harmful Outputs and Ethical Considerations**
>
> We acknowledge the importance of addressing potential harmful outputs and ethical considerations. Ferret-v2 mitigates potential harmful outputs through several mechanisms: it utilizes public, academic datasets for transparency and bias reduction; it integrates fine-grained spatial knowledge to accurately ground descriptions and minimize object hallucinations; and it incorporates instruction-following capabilities from some datasets (e.g., ShareGPT-40k) dataset to handle unanswerable or unclear queries appropriately. We will include a detailed discussion on these points in the revised version.

---

> > ### Comment · Reviewer_NGY1 · 2024-06-04
> > **Response to rebuttal**
> >
> > I would like to thank the authors's response and stick to keeping my rating.

---

> > > ### Author Response · Authors · 2024-06-06
> > > **Thank you for the feedbacks**
> > >
> > > Dear Reviewer NGY1,
> > >
> > > Thank you for your valuable feedback, and we really appreciate the effort and time you took to review our submission. We will address the above concerns in our paper revision.
> > >
> > > Warm regards,
> > >
> > > Author(s)

---

### Decision · Program_Chairs · 2024-07-10

**Decision:**

Accept

**Comment:**

Summary of paper and contributions:
- The paper proposes Ferret-v2, a improved model over Ferret, for vision-language tasks that requires referencing and grounding of regions.  Ferret uses LLM to process image,  language, and region tokens in a unified way.  This work re-examines the Ferret architecture, and introduces three key improvements: 1) handling higher resolution images with the "any resolution" strategy where the images is divided into smaller patches for processing, 2) multi-granular visual encoder where different visual encoders are used for different granularities (CLIP for the global image, and DINOv2 for local patches) 3) three-stage "coarse-to-fine" training process.  The three strategies together leads to an overall method that can work on higher resolution upscaled images and is designed for finer-grained visual-language alignment.  Experiments are conducted on a variety of vision-language tasks (e.g. grounding references, VQA, etc), and shows that Ferret-2 outperforms prior work.  Analysis and ablations are also conducted to justify the various design choices.

Overall Assessment:
- All reviewers are positive on this work.  Some reviewers had questions about whether the work was sufficiently novel (889e) and noted that there could be more discussion about the complexity and scalability of the method, which the authors addressed during rebuttal.  The AC believe the contributions are sufficiently novel, clearly explained and substantiated by the experiments, and agrees the work is of interest to the community and recommends acceptance.

Strengths of the work:
- the proposed designs to be novel and interesting (NGY1,aBcC)
- method is shown to be effective an work well for multiple tasks and benchmarks (NGY1,aBcC,889e)
- paper is well written and easy to follow (NGY1,aBcC)

Recommendations for improvements for the camera ready:
- Add discussion of efficiency and scalability (response to NGY1, aBcC)
- Add discussion of limitations, potential harms and ethical considerations (response to NGY1, aBcC)
- Full pass over paper to improve writing and awkward wording at places.
  - the "any resolution" method should be explained more in the introduction, with appropriate references to earlier work that adopted this
  - For example: 'After devoting to the “any resolution” scaling method' => it is not clear what this means.  Perhaps "With the adoption of the "any resolution" scaling method".